# Effects of Nitrite/Nitrate-Based Accelerators on Strength and Deformation of Cementitious Repair Materials under Low-Temperature Conditions

**DOI:** 10.3390/ma16072632

**Published:** 2023-03-26

**Authors:** Heesup Choi, Masumi Inoue, Hyeonggil Choi, Myungkwan Lim, Jihoon Kim

**Affiliations:** 1Department of Civil and Environmental Engineering, Kitami Institute of Technology, Kitami 090-8507, Japan; 2School of Architecture, Civil Environment and Energy Engineering, Kyungpook National University, Daegu 41566, Republic of Korea; 3School of Architecture, College of Architecture & Design, University of Ulsan, Ulsan 44610, Republic of Korea; 4Faculty of Environmental Technology, Muroran Institute of Technology, Muroran 090-8585, Japan

**Keywords:** cold weather construction, cement-based repair materials, antifreezing agent, NO_2_^−^, NO_3_^−^, strength development, deformation behavior

## Abstract

This study aimed to develop a cementitious repair material that can be constructed in cold weather conditions. The addition of nitrite/nitrate-based antifreezing agents has been shown to increase the initial strength of cementitious repair materials in cold weather. However, increasing the amount of these agents may lead to an increase in deformation behavior and shrinkage cracking. In this study, the effects of different types and amounts of nitrite/nitrate-based antifreezing agents on the strength development and deformation behavior of cementitious repair materials under low-temperature conditions were evaluated. As a result, it was found that the addition of a large amount of calcium nitrite can promote hydration and improve the initial strength of the repair material, irrespective of the type of antifreezing agent. However, this also leads to an increase in shrinkage and the concern of shrinkage cracking. Therefore, a repair material that is repairable in winter was developed by balancing the initial strength and deformation behavior through the appropriate selection of antifreezing agents. The developed repair material can be used to repair structures in cold weather conditions, which is of great significance for the construction industry in Hokkaido, Japan.

## 1. Introduction

Winter construction in Hokkaido, Japan is low, at 19.3% compared to the national average of 34.7% since the amount of snowfall is large and the temperature is lower [1]. When construction is carried out in winter, there are concerns about construction defects such as initial frost damage [2], thus it is necessary to take measures before construction. However, many concrete structures in Japan were constructed during the high economic growth period, and it has been reported that more than 50 years have passed since the construction of the majority of these buildings [3]. Currently, it is thought that repair work will increase in the future owing to the aging of concrete structures. Therefore, there is a need to develop repair materials that can be efficiently used for repair even in winter. The cross-section repair method can be used in combination with other methods, improving the effect of repair [4]. General repair materials that are often used in repair work include polymer cement mortar (PCM), to which a polymer is added to increase the bending and tensile strength, which are weak points of cement-based materials [4]. However, even if this repair material (PCM) is used, if measures are not taken to mitigate issues such as initial frost damage are not taken in winter, it may lead to poor construction and the expected performance may not be exhibited.

Research is being conducted to improve the initial strength and deformation performance of cement repair materials using various materials under low-temperature environments [5,6]. However, examination using cold-resistant accelerators is still not actively being carried out. From previous studies, when a cold-resistant accelerator was added and exposed to a subzero environment, hydration progressed without suffering initial frost damage as the freezing temperature of the kneaded water dropped; it is possible to prevent initial frost damage by producing hydrates from the initial stage and achieving strength development [2,7,8,9,10,11,12]. The cold-resistant accelerators that are presently used are lead and alkaline free and are mainly composed of calcium nitrite (Ca(NO_2_)_2_) and calcium nitrate (Ca(NO_3_)_2_). Additionally, an admixture containing lithium nitrite (LiNO_2_), which is generally used as a rust preventive as a main component, is added as an antifreezing agent, and cases of examining the strength development in a subfreezing environment have also been reported [13]. NO_2_^−^ and NO_3_^−^ in the antifreezing agent play a role in promoting the dissolution of C_3_A, C_3_S, and βC_2_S in cement [14,15]. It has also been reported that C_3_A exhibits increases in the ettringite (3CaO·Al_2_O_3_·3CaSO_4_·32H_2_O): AFt generation speed [16,17] and the additional generation of nitrite/nitrate hydrate (C_3_A/3Ca (NO_2_)_2_/10H_2_O or C_3_A/3Ca(NO_3_)_2_/16–18H_2_O: AFm (NO_2_^−^ or NO_3_^−^)) [14,18,19,20,21]; C_3_S and βC_2_S were reported to exhibit an increase in the production rate of calcium hydroxide (CH) and calcium silicate hydrate (C-S-H) [14,15]. Consequently, the amount of hydrate produced increases as the amount of cold-resistant accelerator added increases at the initial age, and the structure of the cement matrix is densified, which contributes to the enhancement of the initial strength [14,22,23]. However, as the amount of antifreezing agent added increases, the potential for chemical shrinkage of the cement matrix increases. Therefore, reports state that there is a growing concern about the occurrence of shrinkage cracks [24]. According to the “Basics of Cement-based Repair and Reinforcement Materials” of the Japan Cement Association, repair work is conducted with the objective of “restoring the mechanical performance of safety or usability to the extent that the structure possessed at the time of construction” [4]. However, there is a growing concern that cracks will occur after repair work for materials that have large deformation behavior owing to shrinkage when integrating with existing concrete, and this can lead to fatal damage to the structure.

Therefore, when performing repair work with a repair material to which an antifreezing agent has been added in a low-temperature environment, it is necessary to establish control for the reduction of deformation behavior due to shrinkage and the development of good initial strength. Therefore, this study aims to develop a new approach by investigating the effects of different types and amounts of antifreezing agents on the strength development and deformation behavior of cementitious repair materials in a low-temperature environment. Our study will contribute to the development of more effective repair materials and techniques for winter construction in Hokkaido, Japan. Figure 1 shows the study flow chart of this research.

## 2. Materials and Methods

### 2.1. Materials

Table 1 lists the materials used, Table 2 lists the components of the antifreezing agent, and Table 3 shows the chemical composition of the cement used in this experiment. Table 4 shows the mixing ratio of the mortar used that was determined with reference to previous studies [25]. PCM is a material containing organic polymers. However, this research focuses only on inorganic cement-based materials and makes a quantitative evaluation. Therefore, the experiment was conducted using only inorganic materials to which no polymer was added. The water binder ratio was water binder: fine aggregate = 1:1.45, and 18% water was added to the water binder and fine aggregate. In this experiment, both NO_2_^−^ and NO_3_^−^ were treated as having an effect on the promotion of hydration [15]. According to the “Cold-resistant agent operation manual (proposal)” [2], the standard addition amount is 2 to 3% in terms of the solid content with respect to 100 kg of cement. For CN, Ca (NO_2_)_2_ and Ca (NO_3_)_2_ were calculated by the solid content with respect to the binder, and the amount of CN added was 1.5% by mass percent (wt%). The LN was set to 0.8 and 4.2% so that the molar mass was the same as that of the CN anion. CN1 and LN0.8 are added in standard amounts, and CN5 and LN4.2 are added in large amounts assuming the severe winter season. Based on the above, the cases of this experiment were evaluated in five cases of additive-free (N), CN1, CN5, LN0.8, and LN4.2.

### 2.2. Experimental Conditions and Methods

Material management, kneading, placing, and curing were performed in a 5 °C environment according to the “Building Work Standard Specifications/Explanation (JASS 5)” of the Architectural Institute of Japan, which states the condition that “the lower limit of the temperature at the time of unloading can be set to 5 °C” [25,26]. 

The internal temperature history of the mortar was measured immediately after placing it at 24 h of age with a formwork in which a thermocouple was inserted in the center of φ100 mm × 200 mm. 

The compressive strength was measured in accordance with the JIS A 1108 “Method of test for compressive strength of concrete”, where, after kneading, the material was driven into a mold of φ100 mm × 200 mm, and a compressive strength test was performed at a predetermined age (1, 3, and 7 days). The load was applied uniformly so that no impact was applied. The load speed was set to 0.6 ± 0.4 N/mm^2^ per second. The compressive strength was measured three times at each sample level, and the compressive strength results were expressed as the average of these values. 

For the test of autogeneous expansion and shrinkage, we referred to JCI-SAS2-2 “Autogeneous-shrinkage and Autogeneous-expansion test method for cement paste, mortar and concrete (revised edition 2002)” as shown in Figure 2 [27]; a 100 mm × 100 mm × 400 mm iron prism test piece, a 7 mm long styrene board, and a Teflon sheet were used on all surfaces except the placing surface to allow free expansion and contraction. Furthermore, the surface of the specimen was sutured with the Teflon sheet to prevent the evaporation of water. The embedded gauge with temperature measurement was placed at the center of the test piece using a gauge length of 60 mm, a resistance value of 120 Ω, and a coefficient of 2.07%, and the change over time of the strain was measured.

For the test of drying shrinkage, we referred to JIS-A-1129-3 “Methods of measurement for length change of mortar and concrete—Part 3: Method with dial gauge (revised edition 2010)” [28]. We placed the specimens (100 mm × 100 mm × 400 mm) in a controlled environment (20 ± 2 °C and 65 ± 5% relative humidity) and measured the length of the specimens before and after drying for 7 days.

TG/DTG was measured using TG8121 (Thermo plus EVO2 TG-DTA; Rigaku, Tokyo, Japan). The TG/DTG conditions were as follows: measuring temperature = 20~1000 °C, rising speed = 20 °C/min, atmosphere = N_2_-flow, sample weight = 15.00 mg, reference material = α-Al_2_O_3_. The method for preparing the sample used for TG/DTA measurement is as follows: Samples collected at a predetermined age were stop hydrated with acetone, solid-liquid separated by suction filtration, and then dried in an RH11% (relative humidity; RH) environment. A sufficiently dried sample was pulverized to a particle size of 90 μm or less and used for the measurement (see Figure 3). 

MIP (mercury intrusion porosimetry) was measured using Autopore Ⅲ 9400 (Shimadzu, Tokyo, Japan). In this test, as shown in Figure 3, in order to obtain a sample that is representative of the measurement of porosity, samples were collected at each age and cut into 2~5 mm cubes. Then, in order to stop the hydration of the sample, the sample was immersed in acetone for 4 h or more, and dried by the D-dry method for 1 week, and then the pore distribution of each sample was measured at a minimum diameter of 6 nm, with a maximum pressure of 220 MPa.

We focused on the 1st and 7th days of the material age of TG-DTA and MIP and made measurements to consider the effect of the hydrate formation and void structure of the strength development of cementitious repair material and the deformation behavior due to expansion and contraction from the difference in the type and amount of antifreezing agent added.

## 3. Experimental Results and Discussion

### 3.1. Compressive Strength Characteristics

Figure 4 shows the results of the compression test for one day, Figure 5 shows the change over time in the internal temperature history from immediately after placing to 24 h, and Figure 6 shows the change over time in the compressive strength ratio based on N for each material age. 

The compressive strength on day one was 0.9 MPa for N, 1.5 MPa for CN1, 4.2 MPa for CN5, 1.9 MPa for LN0.8, and 3.4 MPa for LN4.2. Compared with N, the compressive strength tended to increase as the amount of antifreezing agent added increased. From the internal temperature history in Figure 5, when adding the antifreezing agent instead of N, AFt and AFm (NO_2_^−^ or NO_3_^−^) are generated by the temperature rise caused by the dissolution of C_3_A at the first peak immediately after kneading due to the hydration reaction of OPC, and CH and C-S-H are generated by the temperature rise [22,23,24,29] caused by the dissolution of C_3_S and βC_2_S, which are considered to be the second peak. Additionally, it is presumed that Al_2_O_3_ contained in the expansion material (3CaO, 3Al_2_O_3_, and CaSO_4_) used in this study was also hydrated by the addition of an antifreezing agent, and the AFt production rate increased [14,22,23,24,26,30]. Compared to N, the initial strength development was observed by adding an antifreezing agent on day one. Particularly, CN5 had a remarkable temperature rise at the first peak, thus it is considered that the highest compressive strength of approximately four times that of N was obtained. When the repair material to which the antifreezing agent is added is exposed to a low-temperature environment, it is considered possible to prevent the initial frost damage by obtaining the initial strength development even after one day. 

From previous studies, it has been reported that when a large amount of CN is added, the strength decreases from three days after the age of the material compared to the control group without CN addition [25]. However, under the conditions of this experiment, no significant decrease in strength was observed at three to seven days even when a large amount of antifreezing agent was added (N is 100%, CN1 is 98%, CN5 is 107%, LN0.8 is 104%, and LN4.2 is 105% at seven days), and a tendency different from that of previous studies was confirmed (see Figure 6) [25]. In this study, a part of the material is replaced with an expansion material. Therefore, it was speculated that the hydration of the expanding material progressed with age and the strength was increased by the continuous generation of the AFt. When the antifreezing agent is added, the amount of AFt and C-S-H produced is larger than that of N at three to seven days of age. Therefore, it is considered that the compressive strength is slightly increased.

### 3.2. Deformation Behavior Characteristics

Figure 7 shows the changes in drying shrinkage strain over time. At one day after the start of drying shrinkage, the drying shrinkage strain was −69 μ for N, −79 μ for CN1, −31 μ for LN0.8, −41 μ for LN4.2, and −122 μ for CN5, indicating that the drying shrinkage tended to increase with the amount of CN and LN added. At seven days after the start of drying shrinkage, the drying shrinkage strain changed at −172 μ for N, −203 μ for CN1, −205 μ for LN0.8, −291 μ for LN4.2, and −477 μ for CN5, indicating that the drying shrinkage observed on day one tended to increase with the amount of CN and LN added, and became more pronounced over time. Moreover, the drying shrinkage strain tended to follow the order CN5 > LN4.2 > LN0.8 ≥ CN1.

These results confirm that drying shrinkage increases with the amount of CN and LN added [25] and that autogeneous shrinkage tests are necessary. In particular, when a significant amount of CN was added (CN5), it was considered that drying shrinkage increased due to the hydration-accelerating effect of CN [22,23,25].

Figure 8 shows the change over time in the autogeneous expansion and shrinkage strain immediately after placing. On day one, N was 111 µ, CN1 was 147 µ, CN5 was −182 µ, LN0.8 was 84 µ, and LN4.2 was 117 µ. Also, at seven days, N was 219 µ, CN1 was 180 µ, CN5 was −214 µ, LN0.8 was 133 µ, and LN4.2 was 200 µ. 

Despite the initial expansion being slightly higher than that of N due to the inclusion of CN or LN in hydration for CN1, LN0.8, and LN4.2, no significant variations were observed until the seventh day. Notably, in all cases, there was a consistent trend towards expanding up to approximately 200 µ. Furthermore, the reason for CN1 exhibiting a slightly greater initial expansion than LN0.8 can be attributed to CN having a larger first peak than LN (see Figure 5). This suggests that the generation of AFt was promoted, leading to an immediate expansion upon contact with water.

However, it was confirmed that CN5 contracted approximately 70 µ in 2 h after contact with water. CN5 has a large effect of promoting hydration and produces a large amount of hydrate (calcium aluminate hydrate of AFt and AFm) from the beginning [14,22,23]. Simultaneously, it is considered that the shrinkage was increased by the relative increase in the amount of C-S-H produced, which significantly affects the shrinkage, from the initial stage of hydration and the increase in the fine voids [31]. 

It has been reported that AFt is produced by the hydration reaction of the expanding material at seven days [32,33]. Therefore, in this study as well, it is considered that the cases excluding CN5 showed expansion due to the influence of AFt. However, CN5, whose contraction increased from the beginning, did not show expansion even at seven days. Therefore, when a large amount of CN is added, there is a concern that shrinkage will increase as the production of C-S-H in the early stage of hydration increases, and care must be taken when calculating the amount to be added [22,23,25].

### 3.3. Quantitative Changes in the Hydrate

Figure 9 shows the results of the TG-DTA measurement after one day, and Figure 10 shows the results of the TG-DTA measurement at seven days. From previous studies, regarding endothermic peaks on the DTG curve, the dehydration of AFt and H_2_O (free water and C-S-H gel water) occurs at approximately 100 °C, decomposition of AFm (NO_2_^−^ or NO_3_^−^) at approximately 200 to 300 °C, CH dehydration at approximately 400 to 500 °C, and the decomposition of C-S-H at approximately 600 to 700 °C [34,35]. 

The mass loss rate of the sample on day one was 11.6% for N, 14.8% for CN1, 18.1% for CN5, 13.5% for LN0.8, and 15.4% for LN4.2. Compared with N, the amount of hydrate produced tended to increase as the amount of antifreezing agent added increased. It was confirmed that the addition of the antifreezing agent promoted the dissolution of C_3_A and the expansion material and increased the amount of AFt produced on the first day of the material’s age. It was also confirmed that AFm (NO_2_^−^ or NO_3_^−^) was generated when a large amount was added compared with the standard addition amount. Previous studies have reported that the addition of a cold-resistant accelerator promotes the hydration of C_3_S and βC_2_S [14,15,22]. However, in this study, no significant difference was observed in the decomposition peaks of C-S-H at approximately 600 to 700 °C. It has been reported that the adsorbed water of C-S-H is dehydrated at approximately 100 °C at the beginning [14,22,23,34,35]. Therefore, it was speculated that the increase in the peak near 100 °C when the antifreezing agent was added compared to N was due to the dehydration of C-S-H along with AFt. It is inferred that the addition of a cold-resistant accelerator increased the amount of AFt and C-S-H produced, whereas the addition of a large amount of a cold-resistant accelerator produced AFm (NO_2_^−^ or NO_3_^−^), resulting in high compressive strength development per day. Particularly, CN5 produces a relatively large amount of C-S-H, which has a greater effect on shrinkage, than in other cases to which a cold-resistant accelerator is added, from the initial stage of hydration, thus it is presumed that the contraction increases while the strength development is obtained [31]. 

At seven days of age, the mass loss rate of the sample was 21.5% for N, 22.5% for CN1, 23.4% for CN5, 21.3% for LN0.8, and 22.0% for LN4.2, showing no significant difference. The amount of C-S-H produced increased with the AFt as the hydration of the expansive material progressed. Therefore, it is presumed that the difference in the compressive strength decreased owing to the increase in the mass reduction rate at approximately 100 °C in all cases (See Figure 6). Previous studies have reported that shrinkage increases with an increase in the amount of the cold-resistant accelerator containing calcium nitrite as the main component under a temperature condition of 10 °C [25]. However, under the conditions of this experiment, the expansion material was replaced, and the formation of AFt was confirmed even at seven days, and it is considered that the contraction was compensated and the tendency to expand.

### 3.4. Changes in Void Structure

Figure 11 and Figure 12 show the void distribution and integrated void distribution at one day, and Figure 13 and Figure 14 show the void distribution and integrated void amount at seven days. Shrinkage is described in “*CONCRETE, Microstructure, Properties, and Materials Second Edition*” by P. Kumar Mehta from the viewpoint of the void structure as follows: “since hydrostatic tension is generated in the water of small capillaries (0.005 to 0.05 µm), removing it causes compressive stress on the solid wall of the capillary voids, causing system contraction” [31]; therefore, we focused on the proportion of voids of 0.05 µm or less. 

As shown in Figure 11 and Figure 12, on day one, in the case excluding CN5, many voids are distributed in 0.5 to 2 µm, and in CN5 only, many voids are distributed in 0.05 to 0.2 µm. Voids of 0.05 µm or less are 11.2% for N, 11.7% for CN1, 32.0% for CN5, 13.9% for LN0.8, and 19.8% for LN4.2, and there was an increasing tendency when an antifreezing agent was added, and it was approximately three times that of the N in CN5. It is considered that when the antifreezing agent was added as compared to N, the voids of 0.5 to 2 µm decreased by promoting the hydration of C_3_A and the expanding material and increasing the amount of AFt produced. Additionally, it is considered that when a large amount of antifreezing agent was added, AFm (NO_2_^−^ or NO_3_^−^) and C-S-H were generated, which further filled the fine voids. It is speculated that CN5, which produces a large amount of hydrate, was filled with voids of 0.5 to 2 µm and shifted to 0.05 to 0.2 µm on the pore side. It is considered that the strength was developed at the age of one day by filling the coarse voids (0.5 to 2 µm) with the increase in the amount of the antifreezing agent added. On the other hand, CN5, along with AFt and nitrite/nitric acid-based hydrate, produces a large amount of C-S-H, which significantly affects shrinkage, from the early stage of hydration, and it is considered that the rapid contraction was caused by an increase in small capillary voids (0.005 to 0.05 µm). At seven days, as shown in Figure 13 and Figure 14, in the case excluding CN5, it became 0.02 to 0.1 µm, and only CN5 became 0.008 to 0.02 µm; the coarse voids were filled with the passage of age, and the main void diameter shifted to the pore side. It was confirmed that voids of 0.05 µm or less tended to have a large amount of CN5, with values of 41.4% for N, 43.6% for CN1, 64.6% for CN5, 43.7% for LN0.8, and 49.0% for LN4.2. AFt was continuously generated by the hydration reaction of the swelling material [32,33], and it is considered that the voids of 0.5 to 2 µm confirmed on day one were significantly reduced. It is presumed that CN5, whose dense void structure was confirmed on day one, was further shifted to the pore side owing to the relative increase in the amount of C-S-H produced with the passage of age. However, the amount of hydrate produced at seven days was not significantly different in all cases. Therefore, it is considered that the difference was small even in the void structure. As such, at seven days, the difference in the void structure was smaller than that of N, even when an antifreezing agent is added. Therefore, it is considered that the compressive strength is relatively the same. However, CN5 shows a tendency to increase voids of 0.03 to 0.01 µm, as shown in the results of TG-DTA mentioned above. It is inferred that the contraction increases as the amount of C-S-H is produced, which significantly affects the contraction and increases relatively from the initial stage of hydration [31].

## 4. Conclusions

In this study, it was assumed that the repair material to which an antifreezing agent is added was exposed to a low-temperature environment immediately after kneading. Experiments were conducted with the aim of clarifying the strength development and expansion–contraction behavior from the viewpoint of quantitative changes in the hydrate and void structure. The findings obtained from this study are summarized below. 

(1)Overall, the results of this study demonstrate that adding an antifreezing agent to repair materials can help mitigate initial frost damage by promoting strength development and filling void structures. The study found that AFm (NO_2_^−^ or NO_3_^−^) is generated, in addition to AFt and C-S-H, when a large amount of antifreezing agent is added, which contributes to initial strength development. Moreover, the strength development was observed to be slightly higher than that of the control group even after three and seven days;(2)However, the addition of antifreezing agents also increases the rate of AFt formation and may accelerate the start of an expansion. In CN5, there is a relative increase in the amount of C-S-H produced from the initial stage of hydration, which can affect shrinkage. Thus, caution must be exercised when calculating the amount of cold-resistant accelerator added. Overall, these findings suggest that antifreezing agents can be a useful tool for improving the durability of repair materials in low-temperature environments.

## Figures and Tables

**Figure 1 materials-16-02632-f001:**
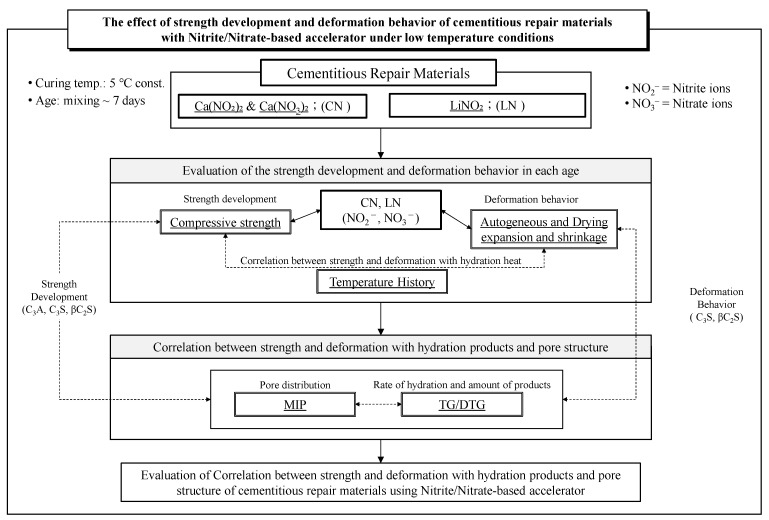
Study Flow Chart.

**Figure 2 materials-16-02632-f002:**
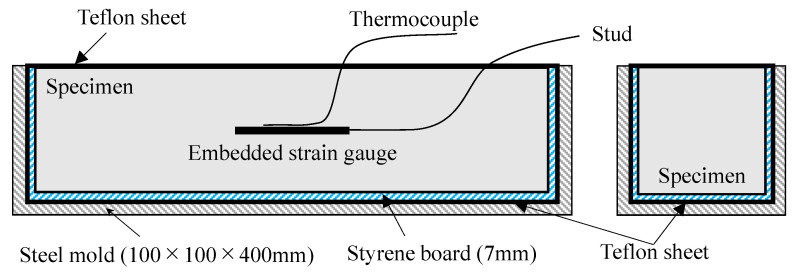
Overview of autogeneous expansion and shrinkage [27].

**Figure 3 materials-16-02632-f003:**
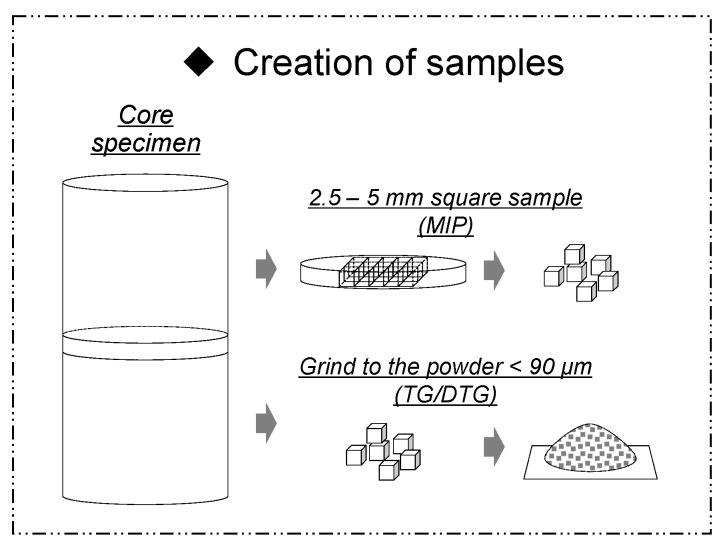
Sampling of specimen for evaluation of MIP&TG-DTA.

**Figure 4 materials-16-02632-f004:**
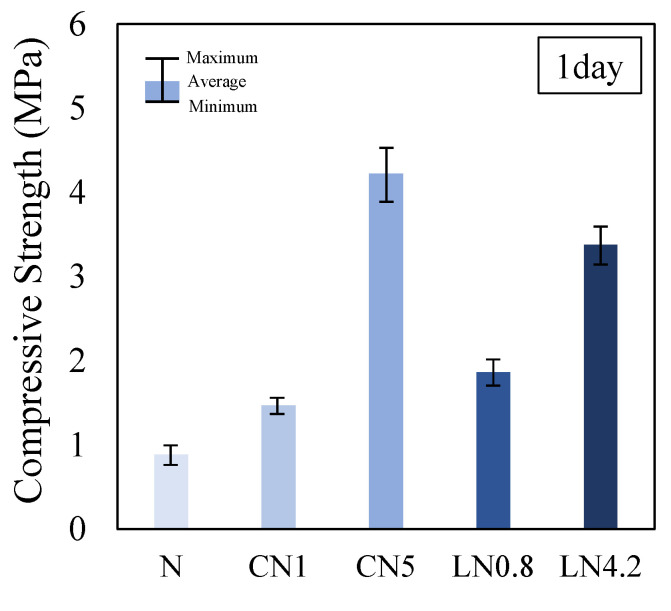
Compressive strength (1 day).

**Figure 5 materials-16-02632-f005:**
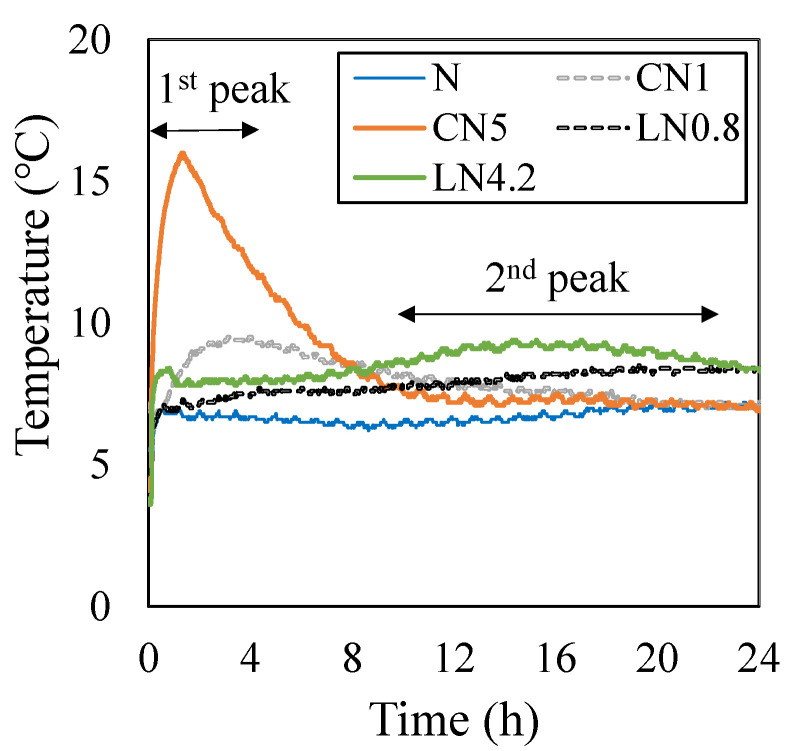
Temperature history (up to 24 h).

**Figure 6 materials-16-02632-f006:**
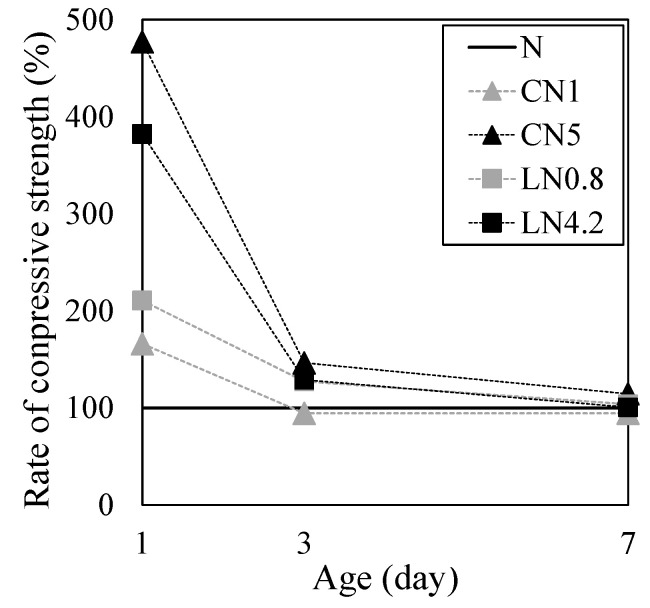
Change of compressive strength ratio of each age.

**Figure 7 materials-16-02632-f007:**
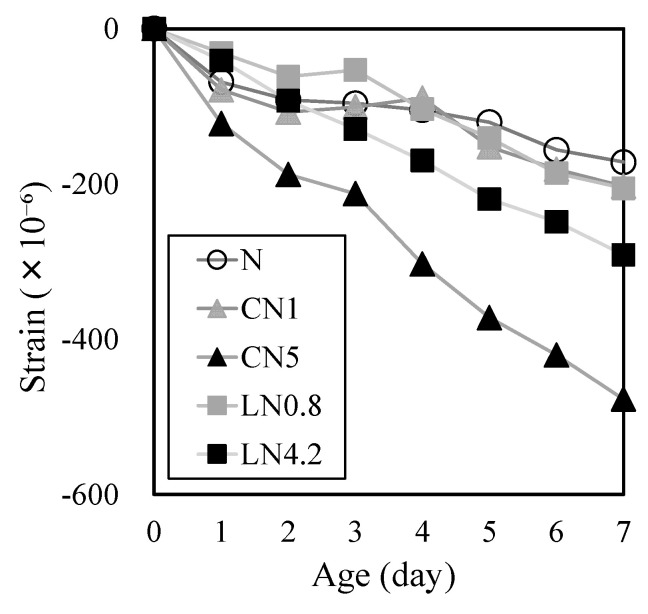
Drying shrinkage (up to 7 days).

**Figure 8 materials-16-02632-f008:**
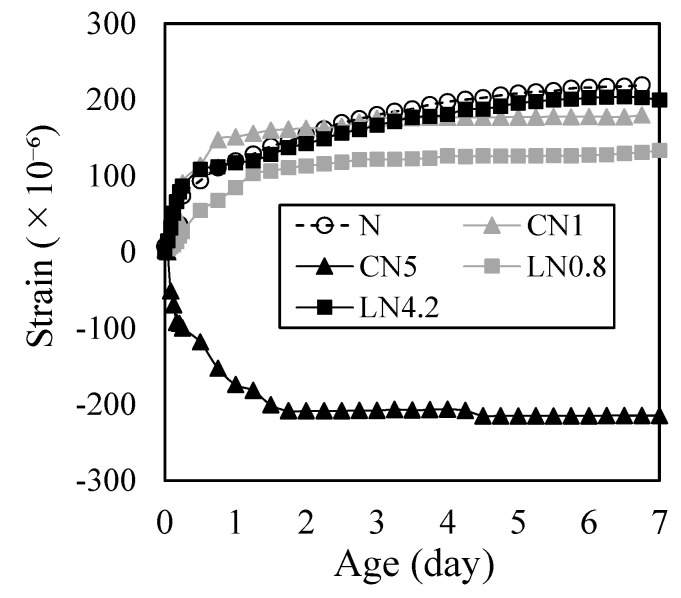
Autogeneous expansion and shrinkage (up to 7 days).

**Figure 9 materials-16-02632-f009:**
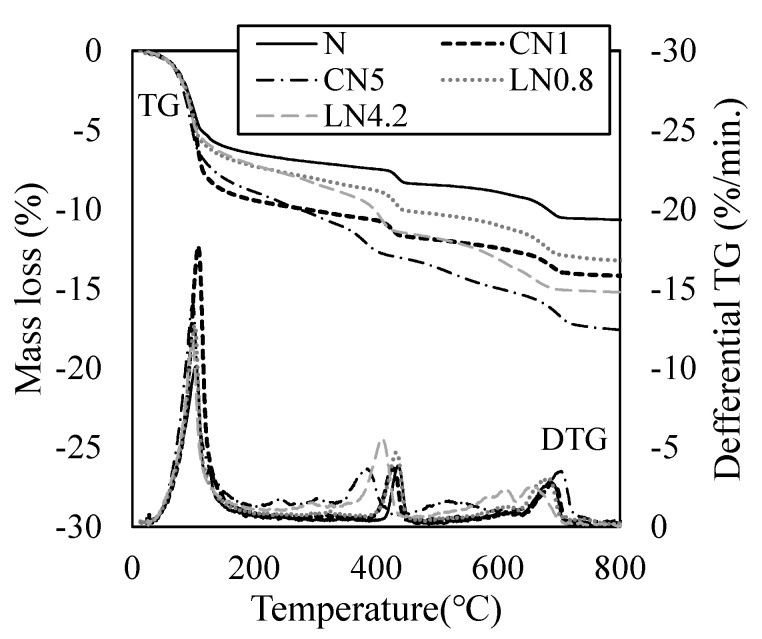
TG/DTG (1 day).

**Figure 10 materials-16-02632-f010:**
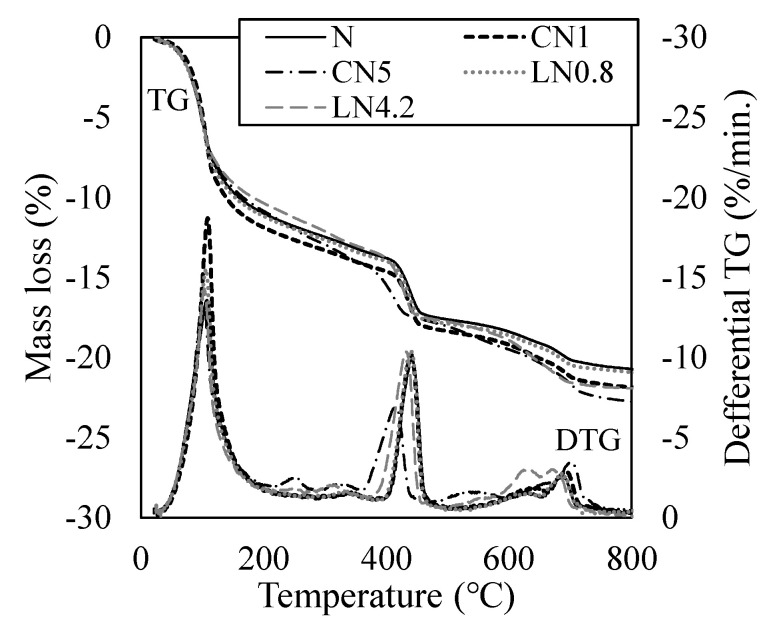
TG/DTG (7 days).

**Figure 11 materials-16-02632-f011:**
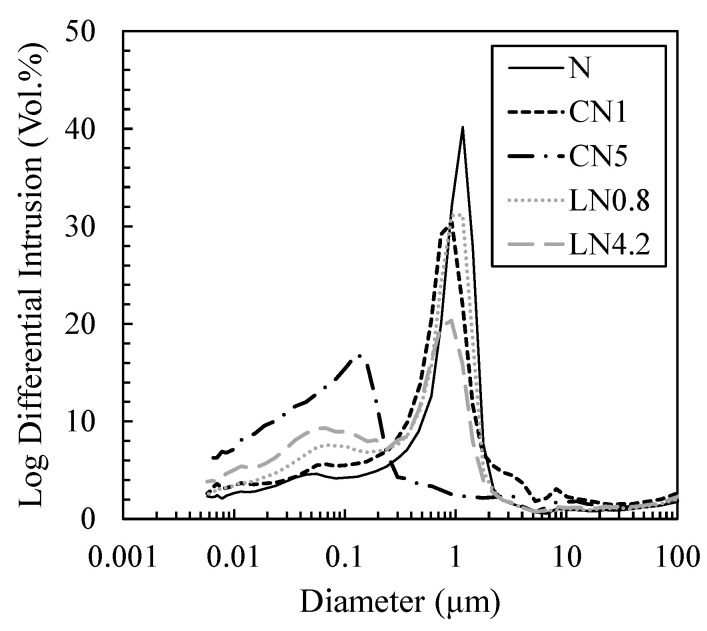
Void size distribution (1 day).

**Figure 12 materials-16-02632-f012:**
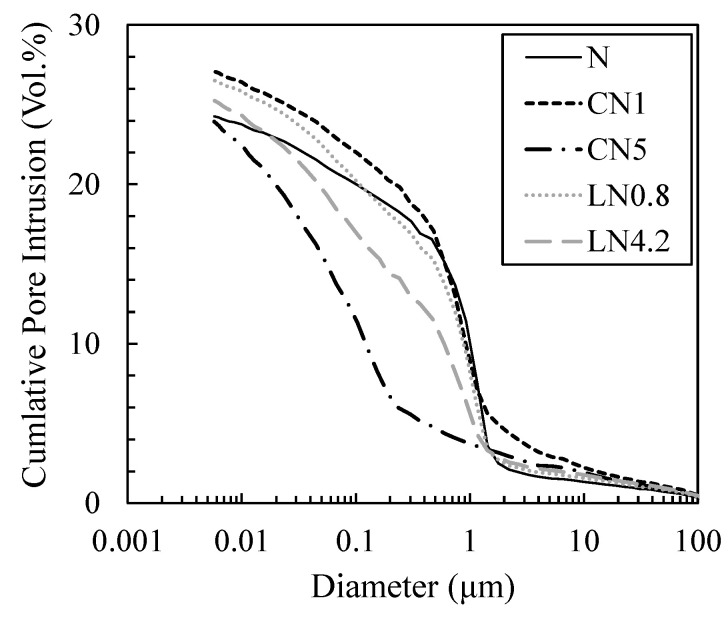
Cumulative void size distribution (1 day).

**Figure 13 materials-16-02632-f013:**
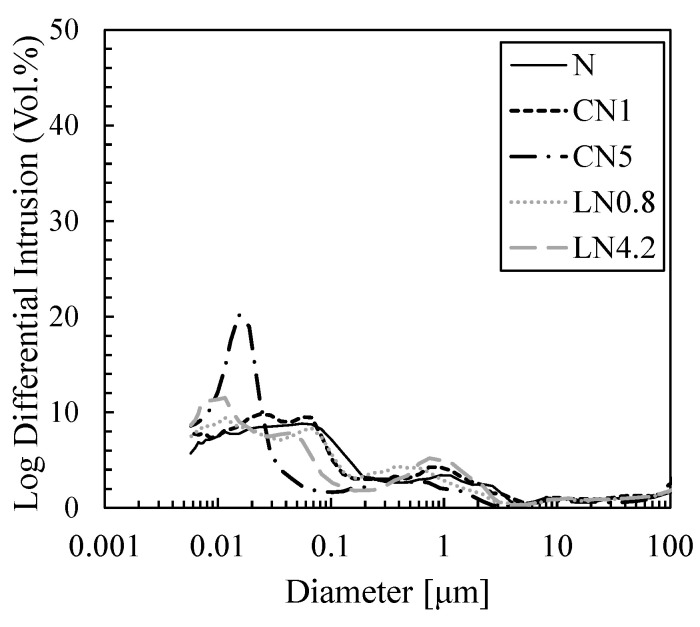
Void size distribution (7 days).

**Figure 14 materials-16-02632-f014:**
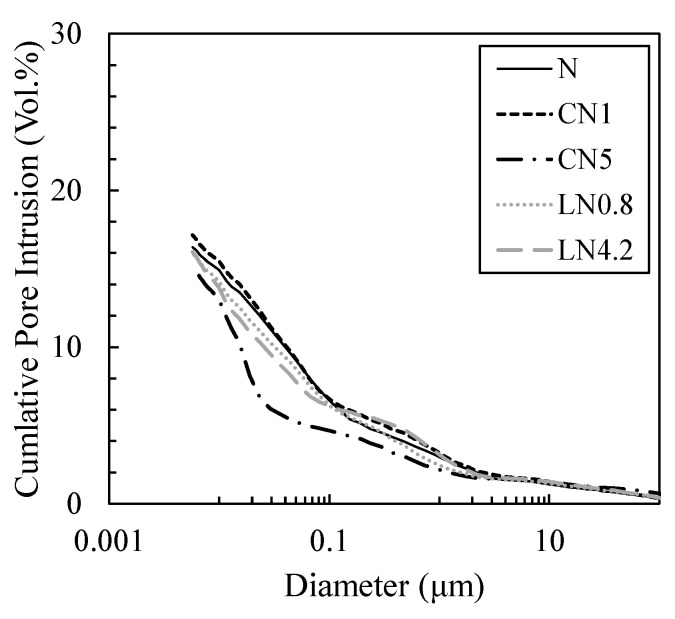
Cumulative void size distribution (7 days).

**Table 1 materials-16-02632-t001:** Properties of the materials.

Materials (Code)		Properties
Cement (C)		Ordinary Portland Cement, Density; 3.16 g/cm^3^
Expansion material (CSA)		Main component; CSA (calcium sulfoaluminate) Expansion Admixture, Density; 2.93 g/cm^3^
Gypsum		Main component; gypsum hemihydrate, Density; 2.64 g/cm^3^
Fine aggregate (S)		No. 5 silica sand, Absolute dry density: 2.61 g/cm^3^, Water absorption: 0.26%, Fineness modulus: 2.16
Antifreezing agent	Calcium Nitrite (CN)	Main component; calcium nitrite, calcium nitrate (45% water solution), Density; 1.43 g/cm^3^
	Lithium Nitrite (LN)	Main component; lithium nitrite (40% water solution), Density; 1.25 g/cm^3^

Note: CN: calcium nitrite = nitrite + nitrate-based accelerator (Ca(NO_2_)_2_ + Ca(NO_3_)_2_); LN: lithium nitrite = nitrite-based accelerator (LiNO_2_).

**Table 2 materials-16-02632-t002:** Properties of the antifreezing agent.

Type	Component	Component Ratio	pH Aquarius Solution	Density of Aquarius Solution (g/cm^3^)
CN	Ca (NO_2_)_2_	23.02	9.3	1.43
	Ca (NO_3_)_2_	22.81		
LN	LiNO_2_	40	9.6	1.25

**Table 3 materials-16-02632-t003:** Chemical composition of cement.

Ordinary Portland Cement	**Chemical Composition (%)**								
	SiO_2_	Al_2_O_3_	Fe_2_O_3_	CaO	MgO	SO_3_	CaSO_4_	Ig.loss	Alkali content
	21.4	5.5	2.8	64.3	2.1	1.9	-	0.56	0.25

**Table 4 materials-16-02632-t004:** Properties of the cement–paste mix.

Type	W/Ⅿ (%)	B:S	Binder (wt%)			Antifreezing Agent (B × wt%)
			Cement	CSA	Gypsum	
N	18	1:1.45	92.3	6.2	1.5	0
CN1						1
CN5						5
LN0.8						0.8
LN4.2						4.2

Note: M: = B + S; B: Binder, S: Sand; CSA: Calcium Sulfoaluminate.

## Data Availability

The data presented in this study are available on reasonable request from the corresponding author.

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
