# Peer review of "Effects of Nitrite/Nitrate-Based Accelerators on Strength and Deformation of Cementitious Repair Materials under Low-Temperature Conditions"

_materials, 2023, doi:10.3390/ma16072632_

Round 1
Reviewer 1 Report (Previous Reviewer 3)
The article entitled "Effects of Nitrite/Nitrate-Based Accelerators on Strength and Deformation of Cementitious Repair Materials under Low Temperature Conditions" submitted for publication on < Materials > journal. My comments are as follows:
1. In Fig 3, XRD and NMR are mentioned without corresponding content in the article.
2. In section 3.1 and 3.2, equipment errors and test errors should be supplemented.
3. Please provide more evidence to further explain how the agents affect internal temperature changes.
4. Language needs to be polished.
5. In introduction part, literature review should be further supplemented to explain the novelty of this work. The novelty of the current version is not fully emphasized.
6. In introduction part, the literature reviews on the cementitious repair material used in cold weather conditions are insufficient, the following references may be helpful.
(1) Characterizing the performance of cementitious partial-depth repair materials in cold climates. Construction and Building Materials 70 (2014) 148–157. DOI: 10.1016/j.conbuildmat.2014.07.114
(2) Ohmic heating curing of carbon fiber/carbon nanofiber synergistically strengthening cement-based composites as repair/reinforcement materials used in ultra-low temperature environment. Composites Part A 125 (2019) 105570. DOI: 10.1016/j.compositesa.2019.105570
Author Response
Answer Sheet
Manuscript ID: materials-2289650
Title: Effects of Nitrite/Nitrate-Based Accelerators on Strength and Deformation of Cementitious Repair Materials under Low-Temperature Conditions
Author's Reply to the Review Report (Reviewer 1)
Comments and Suggestions for Authors:
The article entitled "Effects of Nitrite/Nitrate-Based Accelerators on Strength and Deformation of Cementitious Repair Materials under Low Temperature Conditions" submitted for publication on < Materials > journal. My comments are as follows:
→ We would like to thank the reviewer for carefully reading and for giving quite valuable comments and suggestions, which substantially helped improving the quality of this manuscript. We describe our response point by point to each comment (in bold letters). Our responses are as follows:
Specific comments:
- In Fig 3, XRD and NMR are mentioned without corresponding content in the article.
revised the Fig 3 in the manuscript.
- In section 3.1 and 3.2, equipment errors and test errors should be supplemented.
Reply: In this experiment, we implemented rigorous procedures to minimize the risk of equipment or testing errors. However, we appreciate any detailed feedback you can provide us regarding potential areas for improvement in our testing methodology. Your insights would be invaluable in helping us refine our methods and improve the accuracy and reliability of our results in future experiments.
- Please provide more evidence to further explain how the agents affect internal temperature changes.
revised in the manuscript. And added the references.
- Language needs to be polished.
I did my best to revise the manuscript.
- In introduction part, literature review should be further supplemented to explain the novelty of this work. The novelty of the current version is not fully emphasized.
revised the introduction in the manuscript.
- In introduction part, the literature reviews on the cementitious repair material used in cold weather conditions are insufficient, the following references may be helpful.
(1) Characterizing the performance of cementitious partial-depth repair materials in cold climates. Construction and Building Materials 70 (2014) 148–157. DOI: 10.1016/j.conbuildmat.2014.07.114
(2) Ohmic heating curing of carbon fiber/carbon nanofiber synergistically strengthening cement-based composites as repair/reinforcement materials used in ultra-low temperature environment. Composites Part A 125 (2019) 105570. DOI: 10.1016/j.compositesa.2019.105570
revised in the manuscript. And added the references.
Reviewer 2 Report (New Reviewer)
I want to congratulate the authors for their excellent work; In general, the writing of this manuscript is relatively straightforward and objective, and the results are well presented.
Below I would like to leave some comments that I believe may contribute to the manuscript.
1) At several points in the text, starting with Keywords, the authors present “; NO2, NO3;” to refer to nitrites and nitrates, but it should be remembered that these are anions and therefore have a charge and need to be represented as such. Also, in the Keywords, nitrite is separated by commas, while semicolons separate the others.
2) In the description of the materials, adding additional information, such as brand, purity, and the CAS of the materials used would be interesting. Was the water used distilled or deionized water?
3) In Figure 3, it is mentioned that the material was sprayed to carry out the XRD and NMR techniques. However, these results are not discussed in the work; why?
4) Some discussions, such as the one that begins in line 175 and ends in line 191. They are very interesting but perhaps very speculative since it is attributed that the changes caused by antifreeze promote the formation of certain crystalline phases of the cementitious matrix or promote the dissolution of a specific phase to promote another. How can such statements be verified without the presence of XRD measurements demonstrating the increase or decrease of the mentioned phases?
4) Finally, the authors mention that the measurements of several analyzes were carried out in triplicate. However, none of the figures presented shows whether the data are the mean, much less the standard deviation bar, it would be essential to add such information in the respective figures.
Finally, I would like to thank you again and also congratulate the authors I am in favor of the manuscript being considered for publication after corrections.
Author Response
Answer Sheet
Manuscript ID: materials-2289650
Title: Effects of Nitrite/Nitrate-Based Accelerators on Strength and Deformation of Cementitious Repair Materials under Low-Temperature Conditions
Author's Reply to the Review Report (Reviewer 2)
Comments and Suggestions for Authors:
I want to congratulate the authors for their excellent work; In general, the writing of this manuscript is relatively straightforward and objective, and the results are well presented.
Below I would like to leave some comments that I believe may contribute to the manuscript.
→ We would like to thank the reviewer for carefully reading and for giving quite valuable comments and suggestions, which substantially helped improving the quality of this manuscript. We describe our response point by point to each comment (in bold letters). Our responses are as follows:
Specific comments:
- At several points in the text, starting with Keywords, the authors present “; NO2, NO3;” to refer to nitrites and nitrates, but it should be remembered that these are anions and therefore have a charge and need to be represented as such. Also, in the Keywords, nitrite is separated by commas, while semicolons separate the others.
revised in the manuscript.
- In the description of the materials, adding additional information, such as brand, purity, and the CAS of the materials used would be interesting. Was the water used distilled or deionized water?
Reply: Thank you for the suggestion. While we understand the interest in providing additional information such as brand, purity, and CAS numbers, we can only provide the basic information that was described in the paper at this time. Regarding the water used in the study, it was distilled water. We appreciate the reviewer's understanding on this matter.
- In Figure 3, it is mentioned that the material was sprayed to carry out the XRD and NMR techniques. However, these results are not discussed in the work; why?
revised the Fig 3 in the manuscript.
- Some discussions, such as the one that begins in line 175 and ends in line 191. They are very interesting but perhaps very speculative since it is attributed that the changes caused by antifreeze promote the formation of certain crystalline phases of the cementitious matrix or promote the dissolution of a specific phase to promote another. How can such statements be verified without the presence of XRD measurements demonstrating the increase or decrease of the mentioned phases?
Reply: The authors have previously conducted a study on the strength development and hydration characteristics of ordinary Portland cement with varying amounts of Nitrite/Nitrate-Based Accelerator added (reference 22 in the manuscript). This study aimed to investigate the impact of nitrate and nitrate/nitrite-based accelerators on the strength development and hydrate formation of cementitious materials in cold weather conditions. The results indicated that the nitrate/nitrite-based accelerator exhibited relatively high strength development, even when used alone, and maintained its effectiveness at low temperatures. Furthermore, both nitrate and nitrate/nitrite-based accelerators promoted the initial hydrate formation of cementitious materials. These findings contribute to a better understanding of cementitious materials' strength development and hydrate formation in low-temperature environments and provide information for the design and development of new cementitious materials under such conditions. The study used X-ray diffraction (XRD) to investigate the crystal structure of hydrates in cementitious materials, specifically measuring changes in hydrate structure due to the addition of antifreeze agents in different concentrations. The researchers found that increasing the concentration of antifreeze agents changed the structure of hydrates in cementitious materials and that this structural change was closely related to the strength development of cementitious materials. Based on the results of micro-analyses such as XRD and NMR from the aforementioned study (reference 23), this paper inferred the crystallization of the cementitious matrix due to the addition of antifreeze agents. Additionally, various micro-analysis experiments, including XRD, are currently underway in this study, and efforts will be made to compile and submit additional papers with the results of these experiments in the future.
※reference 22「Yoneyama, A.; Choi, H.S.; Inoue, M.; Kim, J.; Lim, M.; Sudoh, Y. Effect of a Nitrite/Nitrate-Based Accelerator on the Strength Development and Hydrate Formation in Cold-Weather Cementitious Materials. J. Mater. 2021, 14(4), 1006.」
- Finally, the authors mention that the measurements of several analyzes were carried out in triplicate. However, none of the figures presented shows whether the data are the mean, much less the standard deviation bar, it would be essential to add such information in the respective figures.
revised the Fig 4 in the manuscript.
- Finally, I would like to thank you again and also congratulate the authors I am in favor of the manuscript being considered for publication after corrections.
Thanks a lot.
Reviewer 3 Report (New Reviewer)
The Ms. Effects of Nitrite/Nitrate-Based Accelerators on Strength and Deformation of Cementitious Repair Materials under Low Temperature Conditions explored the effects of different types and amounts of nitrite/nitrate-based anti-freezing agents on the strength development and deformation behavior of cementitious repair materials under low temperature conditions.
This paper constitutes an original and well executed work which will be of interest to readers. Furthermore, the language of the manuscript is acceptable and no serious grammatical errors and bad sentence structure were noted. However, the authors barely explain the results and the effects of Nitrite/Nitrate-based accelerators on strength and deformation, which is a leading weakness in the manuscript scientific contents. The relation between reaction process and mechanism of cementitious repair materials under low temperature conditions were not analyzed in depth, and lacked quantitative analysis and evaluation.
Therefore, the authors should give more analysis and exploration on reaction process and mechanism. From above reason, I would recommend the publication of this manuscript in the excellent journal after conducting a major change.
Author Response
Answer Sheet
Manuscript ID: materials-2289650
Title: Effects of Nitrite/Nitrate-Based Accelerators on Strength and Deformation of Cementitious Repair Materials under Low-Temperature Conditions
Author's Reply to the Review Report (Reviewer 3)
Comments and Suggestions for Authors:
The Ms. Effects of Nitrite/Nitrate-Based Accelerators on Strength and Deformation of Cementitious Repair Materials under Low Temperature Conditions explored the effects of different types and amounts of nitrite/nitrate-based anti-freezing agents on the strength development and deformation behavior of cementitious repair materials under low temperature conditions.
This paper constitutes an original and well executed work which will be of interest to readers. Furthermore, the language of the manuscript is acceptable and no serious grammatical errors and bad sentence structure were noted. However, the authors barely explain the results and the effects of Nitrite/Nitrate-based accelerators on strength and deformation, which is a leading weakness in the manuscript scientific contents. The relation between reaction process and mechanism of cementitious repair materials under low temperature conditions were not analyzed in depth, and lacked quantitative analysis and evaluation.
Therefore, the authors should give more analysis and exploration on reaction process and mechanism. From above reason, I would recommend the publication of this manuscript in the excellent journal after conducting a major change.
→ We would like to thank the reviewer for carefully reading and for giving quite valuable comments and suggestions, which substantially helped improving the quality of this manuscript. We describe our response point by point to each comment (in bold letters). Our responses are as follows:
Reply: In response to the reviewer's feedback, we have added content on the reaction mechanisms and references related to cement-based materials mixed with antifreeze agents, and have made necessary corrections.
As the reviewer may be aware, calcium-nitrite and calcium-nitrate are increasingly being used as the main components of chloride- and alkali-free anti-freezing agents for cold weather concreting. These nitrite/nitrate-based accelerators can accelerate the hydration of tricalcium aluminate (C3A phase) and tricalcium silicate (C3S phase) in cement, leading to improved early strength and effective prevention of frost damage. In areas with temperatures below -10°C, larger amounts of these accelerators are often used. However, the relationship between hydration processes and strength development in concrete with significant amounts of nitrite/nitrate-based accelerators remains unclear.
We previously conducted a study on the strength development and hydration characteristics of ordinary Portland cement with varying amounts of Nitrite/Nitrate-Based Accelerator added (references 22, 23, 25, 29 in the manuscript) over several years. The aim of these studies was to investigate the impact of nitrate and nitrate/nitrite-based accelerators on the strength development and hydrate formation of cementitious materials under cold weather conditions. The results indicated that nitrate/nitrite-based accelerators exhibited relatively high strength development, even when used alone, and maintained their effectiveness at low temperatures. Furthermore, both nitrate and nitrate/nitrite-based accelerators promoted the initial hydrate formation of cementitious materials. These findings contribute to a better understanding of the strength development and hydrate formation of cementitious materials in low-temperature environments, and provide useful information for designing and developing new cementitious materials for such conditions.
Notably, in reference 22, we used techniques such as X-ray diffraction (XRD), Solid-State Nuclear Magnetic Resonance (NMR), Scanning Electron Microscope (SEM), and Thermogravimetric Differential Thermal Analysis (TG/DTG) to investigate the crystal structure of hydrates in cementitious materials. Specifically, we measured changes in hydrate structure due to the addition of antifreeze agents in different concentrations. We found that increasing the concentration of antifreeze agents changed the structure of hydrates in cementitious materials, and that this structural change was closely related to the strength development of cementitious materials.
In addition, we are currently conducting various micro-analysis experiments, including XRD, NMR, and SEM, and we plan to compile and submit additional papers with the results of these experiments in the future.
Reviewer 4 Report (New Reviewer)
Congratulations to the authors of the article . It is complete and they have shown great knowledge in the field of cement hydration. The reviewer hopes that the authors will develop this field of research
Author Response
Answer Sheet
Manuscript ID: materials-2289650
Title: Effects of Nitrite/Nitrate-Based Accelerators on Strength and Deformation of Cementitious Repair Materials under Low-Temperature Conditions
Author's Reply to the Review Report (Reviewer 4)
Comments and Suggestions for Authors:
Congratulations to the authors of the article . It is complete and they have shown great knowledge in the field of cement hydration. The reviewer hopes that the authors will develop this field of research
Reply: Thanks a lot.
Round 2
Reviewer 1 Report (Previous Reviewer 3)
it can be accepted with this revised form.
Reviewer 3 Report (New Reviewer)
The author responded to my concern although no corresponding improvememts had been made. But, I think the explanation given by the authors caould be acceptabled. So, I agree accept this paper in present form.
This manuscript is a resubmission of an earlier submission. The following is a list of the peer review reports and author responses from that submission.
Round 1
Reviewer 1 Report
In this paper, the effects of nitrite/nitrate accelerators on the strength development and deformation behavior of cement-based repair materials at low temperature were investigated. There are many problems in this paper. Here are the detailed comments:
- In Abstract, the author described the research background too much in the abstract, but did not mention the innovation and main research results of this paper. The author should rewrite it.
- Line 38: Ref [1], The data cited by the author comes from 2008, and now it has been 14 years. The author should quote the data of recent three years to ensure its timeliness and accuracy.
- Line 45-46: “The cross-section repair method can be used in combination with other methods, improving the effect of repair .” The author please expand on this point of view, not just a simple literature citation.
- Line 49-52: “ ...even if this repair material (PCM) is used, if measures such as initial frost damage are not taken in winter, it may lead to poor construction and the expected performance may not be exhibited.” At present, there are many literatures about self-healing concrete. Many researchers have found that even under the severe cold conditions, the repair materials will not affect the concrete construction and can obtain a good self-healing effect. Where did the author get the above views? Please indicate the source.
- The introduction does not explain the innovation of this paper, so we can't see the research significance of this paper. The references cited in this article are too old, and the author needs to re-cite the literature of the recent year, which is also an aspect of the novelty of the paper. In addition, many of the author's views in the introduction are lack of scientific basis, it is suggested that the author rewrite the introduction.
- Table 1, Ordinary Portland Cement, 32.5, 42.5 or 52.5?
- In Section 3.1, it may be better to change the unit "N/mm2" to "MPa".
- Line 170-172: “Compared with N, the......anti-freezing agent added increased.” what’s the reason?
- Line 173: “AFt and AFm (NO2 or NO3)...” What is the relationship between the two?
- After 7 days, the strength of CN1 is lower than that of N, and the strength of LN is the same as that of N. Does it mean that the added admixture has not played the expected effect. The author can not only describe the experimental results, but also need in-depth analysis. Section 3.1 is difficult to read and is recommended to be rewritten.
- Figures 6-9 are not clear, it is recommended to redraw them.
- Figure 10 shows that a large number of pore sizes of samples which are above 0.1 μm, indicating bad mechanical properties.
Reviewer 2 Report
I have read the Manuscript entitled " Effect of strength development and deformation behavior of cementitious repair materials with Nitrite/Nitrate-based accelerator under low temperature conditions" and found it very interesting.
This work focuses on the strength development and deformation behavior of cementitious repair material added with anti-freezing agent. However, some minor modifications are required. My specific comments are as follows:
- In the introduction, more previous studies on deformation of cement-based material should be discussed and more reference should be added, for example “Influence of reactivity and dosage of MgO expansive agent on shrinkage and crack resistance of face slab concrete”.
- The format of the chemical formula is not uniform in figure 1. Authors should check the whole figure and make corrections.
- It is noted that the tense in this manuscript is inconsistent. Authors should carefully check the whole manuscript.
- In lines 131-132 and figure 2, authors should add references support.
- In 3.2. Deformation behavior characteristics, the sample CN5 shows a completely different phenomenon from others, and the author is better to cite relevant reference here validate these phenomena and explanations.
- The void structure of cement-based materials can also be discussed in 3.4. Changes in void structure. Here are some articles for your reference, including “Fractal analysis on pore structure and hydration of magnesium oxysulfate cements by first principle, thermodynamic and microstructure-based methods” , “The influence of fiber type and length on the cracking resistance, durability and pore structure of face slab concrete”.
- There is lack of explanation of void structure change of CN5 in 3.4. Changes in void structure.
- There are many mistakes in the format of reference. Please check and make corrections.
Reviewer 3 Report
In this work, a repair material was developed and the strength development and deformation behavior with changes in the type and amount of nitrite/nitrate-based anti-freezing agent were thoroughly investigated. I think this work is with high quality and after a tiny revision it can be published. Detailed suggestions are as below:
1. The unit for compressive strength in Figure. 4 needs to be revised to promise the accurate exprestion.
2. The expression style for TG/DTA and MIP analyses needs to be changed, the existing one is not obvious and can't promise the direct reflection of the test results. Following literatures may be helpful.
1) Multi-structural evolution of conductive reactive powder concrete
manufactured by enhanced ohmic heating curing
2) Ohmic heating curing of high content fly ash blended cement-based composites towards sustainable green construction materials used in severe cold region
3. The language in Conclusion part needs to be fully checked, some expression is kind of confusing and not readable.
Reviewer 4 Report
I have finished reviewing the manuscript Ref. No: materials-1721751 Titled " Effect of strength development and deformation behavior of cementitious repair materials with Nitrite/Nitrate-based accelerator under low temperature conditions", submitted toMaterials
In general, I believe that the topic, which is presented in the paper, is suitable for publishing in the present journal. This paper can be accepted for publication.
Reviewer 5 Report
The manuscript materials-1721751 presents an experimental assessment of the effects of accelerators containing nitrites and nitrates on the strength development and shrinkage behaviour of cementitious mixes in a cold environment (5°C).
Degree of novelty: unfortunately, this Reviewer could not identify the real novelty brought in this contribution. Several studies have been conducted in similar directions, and the conclusions here depicted are quite easily predictable. Surprisingly, the 4 (!) papers https://doi.org/10.3390/ma12233936, https://doi.org/10.3390/ma13173686, https://doi.org/10.3390/ma12172706, https://doi.org/10.3390/ma14041006 by the same group of authors (and published in the same journal over the last 3 years) overlap at a notable extent with the contents of the current manuscript.
Introduction: the introduction is very confused and superficial, and fails to frame the problem properly.
Language and style: the quality and the readability of the paper are poor, and unacceptable for the requirements of a reputed scientific journal. Sentences are often confused and hardly understandable.
Additional remarks: the Authors attempt to describe the behaviour of mortars in the framework of Hokkaido, where the average temperature in winter time can reach -10°C. Here, tests are conducted at 5°C, which is indeed a low temperature, however, it is very far from the context depicted by the Authors in the Introduction. Despite I understand that specific guidelines were followed, the authors should clarify the validity in the practice of the presented results, in the context of this specific area of Japan.
Besides, mechanical tests are conducted with (only) 3 repetitions, and the statistical variability of the results is not shown and discussed, strongly impairing the rigour of the paper.
Unfortunately, the scientific rigour and the innovation of the manuscript are far from complying with the standards of the journal.